# Runx1 shapes the chromatin landscape via a cascade of direct and indirect targets

**Matthew R. Hass**[1,2], **Daniel Brissette**[1,2], **Sreeja Parameswaran**[1,3], **Mario Pujato**[1,3], **Omer Donmez**[1,3], **Leah C. Kottyan**[1,2,3], **Matthew T. Weirauch**[1,2,3,4]*, **Raphael Kopan**[1,2]*

**1** Department of Pediatrics, University of Cincinnati College of Medicine, Cincinnati, Ohio, United States of America, **2** Division of Developmental Biology, Cincinnati Children's Hospital Medical Center, Cincinnati, Ohio, United States of America, **3** Center for Autoimmune Genomics and Etiology, Cincinnati Children's Hospital Medical Center, Cincinnati, Ohio, United States of America, **4** Division of Biomedical Informatics, Cincinnati Children's Hospital Medical Center, Cincinnati, Ohio, United States of America

* Matthew.Weirauch@cchmc.org (MTW); Raphael.Kopan@cchmc.org (RK)

## Abstract

Runt-related transcription factor 1 (Runx1) can act as both an activator and a repressor. Here we show that CRISPR-mediated deletion of *Runx1* in mouse metanephric mesenchyme-derived mK4 cells results in large-scale genome-wide changes to chromatin accessibility and gene expression. Open chromatin regions near down-regulated loci enriched for Runx sites in mK4 cells lose chromatin accessibility in *Runx1* knockout cells, despite remaining Runx2-bound. Unexpectedly, regions near upregulated genes are depleted of Runx sites and are instead enriched for Zeb transcription factor binding sites. Re-expressing Zeb2 in *Runx1* knockout cells restores suppression, and CRISPR mediated deletion of Zeb1 and Zeb2 phenocopies the gained expression and chromatin accessibility changes seen in Runx1KO due in part to subsequent activation of factors like Grhl2. These data confirm that Runx1 activity is uniquely needed to maintain open chromatin at many loci, and demonstrate that Zeb proteins are required and sufficient to maintain Runx1-dependent genome-scale repression.

**Data Availability Statement:** All the sequencing data has been deposited at the Gene Expression Omnibus (GEO) and can be located under accession number GSE158093.

## Author summary

Runt-related transcription factor (Runx) 1 & 2 impact development and disease by activating or repressing transcription. In this manuscript we used genome editing tools to remove Runx1, and as expected, observed widespread changes in chromatin accessibility. Newly closed areas contained Runx1 binding sites and were enriched near genes whose expression depended on Runx1. Interestingly, this occurred despite continued binding of Runx2 to the same regions of DNA, which suggests that Runx2 is insufficient to maintain open chromatin and expression of Runx1 target genes in this cellular context. By contrast, newly opened chromatin regions, many near genes that were upregulated in Runx1 knockout cells, did not enrich for Runx1 binding sites. Instead, these regions were enriched for sites for the repressor Zeb proteins. We found that the loss of Zeb 1 & 2 expression, direct transcriptional targets of Runx1, resulted in the opening of chromatin

**Funding:** This work was made possible through funding from the following sources: R01 GM055479 to R.K. and M.T.W., R01 NS099068 to M.T.W. and Cincinnati Children's Hospital 'Trustee Award', 'Center for Pediatric Genomics Award' and 'CCRF Endowed Scholar Award' to M.T.W. The funders had no role in study design, data collection and analysis, decision to publish, or preparation of the manuscript.

**Competing interests:** The authors have declared that no competing interests exist.

and upregulation of genes residing near the newly open sites in Runx1 knockout cells. The same sites were also open and nearby genes expressed in edited Zeb1 and Zeb2 knockout cells. Among them were transcription factors, such as the Grhl2 gene, which in turn bind to and upregulate their target genes. Thus, the loss of a single transcription factor initiates a cascade of direct and indirect ramifications with likely negative effects on development and health.

## Introduction

Mammalian genomes encode over 1,000 Transcription factors (TFs) which precisely control gene expression through complex combinatorial interactions and transcriptional cascades [1], executing the first step in translating genomic DNA sequence into function. To achieve this precision, TFs use a wide range of mechanisms, including initiating the activation or repression of gene expression directly or through the recruitment of co-factors, initiating new chromatin looping interactions between enhancers and target promoters, and altering the chromatin landscape through the repositioning of nucleosomes [2]. Achieving an understanding of the mechanisms underlying precise control of gene expression is thus an enduring and fundamental goal of molecular biology.

Runx/Runt TF family members recognize a characteristic TGTGGT DNA-binding motif, are conserved across all metazoans [3], and play important roles in development and disease [4,5], notably during hematopoiesis [6,7], skin development [8–10], and ossification [4,11–13]. Runx proteins can act as repressors, by recruiting the Groucho/TLE proteins via a C-terminal tetrapeptide WRPY, or as activators, by heterodimerizing with Core binding factor (CBF)ß and recruiting cell context-specific activators. Although the Runx proteins often play redundant roles when co-expressed given their homologous protein structures, post-translational modification-dependent co-factor interactions enable unique functions of specific Runx proteins [14–16]. In several developmental contexts, Runx proteins collaborate with Notch, at times facilitating Notch activity [17,18], and at others acting downstream of Notch [19]. The role that Runx1 plays in establishing chromatin accessibility has been studied in some detail within the hematopoietic system [20], where it acts to maintain chromatin accessibility. However, how Runx proteins influence gene regulatory networks in different cellular contexts remains to be elucidated.

We used the mouse metanephric mesenchyme-derived mK4 cells because they resemble cells undergoing mesenchymal to epithelial transition [21], which is a process critical to kidney development. Previously, we found that Runx binding sites were enriched near Notch-bound enhancers in mk4 cells [22]. Two of the three Runx paralogs, Runx1 and Runx2, are expressed in mK4 cells [21], facilitating detailed molecular comparison of Runx1 versus Runx2 functions in regulating gene expression and their integration with multiple signaling pathways. Such analyses are further aided in mK4 cells by the normal karyotype, the ease of CRISPR-mediated genetic manipulation, and by short doubling times, providing sufficient material for a variety of genomic assays.

In this study, we show that Runx1 plays an important role in regulating chromatin accessibility at many genomic loci in mK4 cells. In the absence of Runx1, Runx2 bound most of the Runx1-bound chromatin but could not maintain Runx1-dependent accessibility or gene expression. As Runx1 can repress expression of some genes, we anticipated re-expressed genes in Runx1KO cells to be actively repressed by Runx1; however, we were surprised to discover that accessible chromatin near loci expressed only after Runx1 deletion were depleted of Runx

sites and instead enriched for Zeb sites, suggesting indirect involvement of Runx1 in their regulation. Further investigation revealed that repression at multiple loci throughout the genome is mediated by two Runx1-dependent targets, Zeb1 and Zeb2. Zeb proteins are sufficient for repression since restoring Zeb2-expression in Runx1KO cells repressed ectopically expressed genes in Runx1KO cells. We demonstrated that Zeb repressors were required for repression in the presence of Runx1 by inactivating Zeb1 and Zeb2 in parental mK4 cells, which phenocopied the gained accessibility and expression seen in Runx1 knockout. Thus, the direct impact of Runx1 on chromatin in mK4 cells is mediated primarily through its ability to promote and maintain chromatin accessibility and transcriptional activator function, rather than through its repressor function. Collectively, these data provide mechanistic insight into how Runx and Zeb TFs interactively control gene expression and place Runx1 at a top of a TF cascade maintaining the global chromatin landscape.

## Results

### Generation and characterization of Runx1 knockout cells

To generate cells lacking Runx1 activity, we targeted Runx1 with two gRNAs flanking exon 3, which contains the start codon of the transcript expressed in mK4 cells and encodes part of the Runt DNA binding domain (Fig 1A). Multiple clones grew after selection with puromycin showing deletion of the targeted exon 3 (Fig 1A) by PCR genotyping and loss of Runx1 protein as confirmed by Western blot (Fig 1B). The expression of Runx2 in these cells remained unchanged (Figs 1B and S1). Thus, any functional differences between Runx1KO cells and the parental mK4 cells (control) would indicate potential Runx1-specific roles that cannot be compensated for by Runx2.

To determine the impact of Runx1-deficiency, we performed multiple genomic assays comparing Runx1KO to parental mK4 cells. Specifically, we analyzed gene expression through RNA-seq, identified genomic locations bound by Runx1 and Runx2 through ChIP-seq, and mapped chromatin architecture by ATAC-seq (Fig 1C). The RNA-seq, ChIP-seq, and ATAC-seq experiments were all performed in biological triplicates to enable statistical analyses for the identification of significant differences between control and Runx1KO cells.

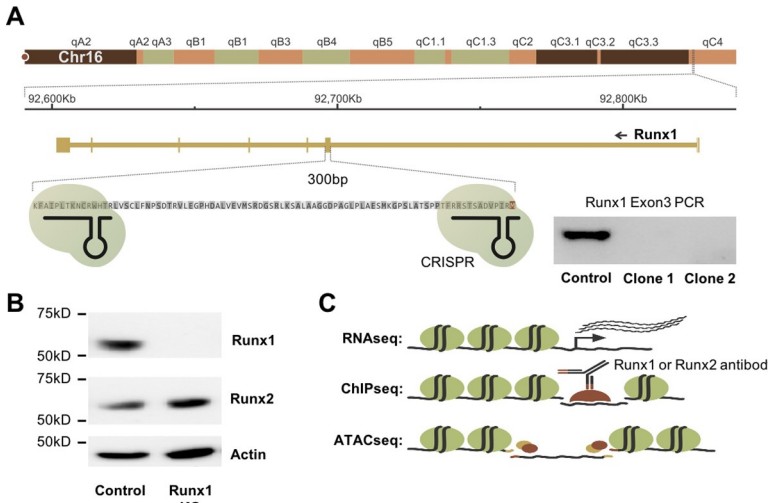

**Fig 1. Generation and Characterization of Runx1KO Cells.** A) Diagram of the Runx1 exon 3 region targeted for deletion using CRISPR-Cas9 and confirmation of deletion by PCR. B) Western blot showing that Runx1KO cells lack Runx1 protein but contain Runx2. C) Schematic of genomic analyses utilized to characterize Runx1KO cells.

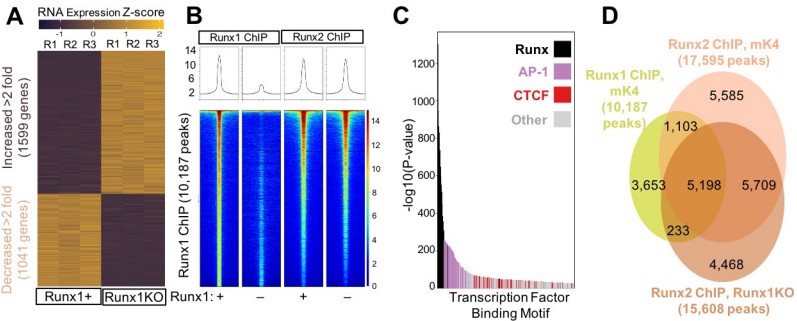

**Fig 2. Widespread Transcriptional Changes in Runx1KO Cells Despite Runx2 Largely Occupying the Same Regions as Runx1.** A) Heatmap of RNA-seq triplicates showing 1,705 upregulated and 1,182 downregulated genes (over 2 fold) in Runx1KO cells compared to control mK4 cells. B) Heatmaps of ChIP-seq reads mapping to Runx1 peaks from Runx1 ChIP or Runx2 ChIP in control versus Runx1KO cells with all four heatmaps being the same Runx1 ChIP peak locations ordered by the strength of the Runx1 ChIP peak. C) Graph displaying -log10 p-values of motif enrichment, revealing that Runx1 motifs are the most highly enriched motifs in the Runx1 ChIP peaks. D) Venn diagram showing the overlap of Runx1 and Runx2 ChIP peaks. Note, the lower overlap of called peaks in the Venn diagram is likely due to the Runx2 signal being just below the threshold for peak calling for many of the peaks on the lower half of the heatmap in A. RELI analysis confirmed the very significant overlap of the Runx1 and Runx2 ChIP (138.5 fold enrichment, corrected p-value 1.45 x 10$^{-218}$).

## Runx1-deficiency induces dramatic changes in gene expression

RNA-seq analysis identified 1,599 upregulated and 1,041 down-regulated transcripts that were significantly altered across highly consistent replicates in Runx1KO cells relative to control cells (Fig 2A; fold change > 2-fold, FDR < 0.05; S1 Table). Consistent with previous studies, Runx1KO cell downregulated genes were enriched for the GO term identifying TGF-beta receptor signaling pathway (23.73 fold enriched, p-value 0.0012, FDR 0.039 S2 Table) [23]. analysis of the upregulated genes in Runx1KO cells identified enrichment for biological processes including antigen processing and presentation (6.38 fold enriched, p-value 5.83 x 10$^{-07}$, FDR 2.3 10$^{-04}$, Tables 1 and S2), consistent with the critical role that Runx1 plays in the immune system and with observations in human patients with Runx1 mutations [24]. The widespread changes in gene expression caused by Runx1-deficiency are consistent with

**Table 1. Gene Ontology of Runx1 Regulated Genes.** This table shows the top 5 gene ontology categories from the S2 Table, based on the fold enrichment for genes down-regulated in Runx1KO cells (top) or down-regulated in Runx1KO cells (bottom).

| Runx1KO Down-regulated Gene Biological Process Gene Ontology | GO number | Fold Enrichment | p-value | FDR |
|---|---|---|---|---|
| Transforming growth factor beta receptor complex assembly | 7181 | 23.73 | 1.20x10$^{-03}$ | 3.39x10$^{-02}$ |
| Condensed mesenchymal cell proliferation | 72137 | 17.8 | 2.04x10$^{-03}$ | 4.99x10$^{-02}$ |
| Regulation of serotonin uptake | 51611 | 17.8 | 2.04x10$^{-03}$ | 4.98x10$^{-02}$ |
| Chemoattraction of axon | 61642 | 17.8 | 2.04x10$^{-03}$ | 4.98x10$^{-02}$ |
| Negative regulation of synapse assembly | 51964 | 15.82 | 4.59x10$^{-04}$ | 1.63x10$^{-02}$ |
| **Runx1KO Up-regulated Gene Biological Process Gene Ontology** | **GO number** | **Fold Enrichment** | **p-value** | **FDR** |
| Positive regulation of cGMP-mediated signaling | 10753 | 9.3 | 2.32x10$^{-04}$ | 2.16x10$^{-02}$ |
| Gamma-delta T cell activation | 46629 | 8.45 | 3.40x10$^{-04}$ | 2.85x10$^{-02}$ |
| Regulation of ribonuclease activity | 60700 | 8.26 | 3.86x10$^{-05}$ | 5.38x10$^{-03}$ |
| Positive regulation of epidermal growth factor-activated receptor activity | 45741 | 7.15 | 6.72x10$^{-04}$ | 4.72x10$^{-02}$ |
| Antigen processing and presentation of endogenous peptide antigen via MHC class I via ER pathway, TAP-independent | 2486 | 6.38 | 5.83x10$^{-07}$ | 2.30x10$^{-04}$ |

previous observations of non-redundancy with Runx2, as seen in other cellular contexts [4]. Thus, Runx1 plays a critical role in controlling the transcriptome within mK4 cells in a manner that cannot be compensated for by Runx2. We next examined if differences between Runx1 and Runx2 effects on gene expression might be due to differences in DNA binding preferences or differences in genomic binding locations.

## Runx1 and Runx2 bind near genes that are down-regulated in Runx1KO cells

To identify genomic loci occupied by Runx proteins, we performed Runx1 and Runx2 ChIP-seq. ChIP-seq in mK4 cells produced 10,187 peaks that were highly reproducible between replicates but not present in Runx1KO cells, illustrated in heatmaps of the Runx1 ChIP-seq reads mapped to peaks in control and Runx1KO cells (Figs 2B and S2A) and in the correlation matrix of the reads on the Runx1 ChIP peaks between the various samples (S2B Fig). HOMER transcription factor binding site motif enrichment analysis using mouse motifs from the Cis-BP database including all reported motifs for various TFs [25] identified Runx motifs as the most highly enriched in the control cell dataset (p-value $1 \times 10^{-1298}$) (Fig 2C and S3 Table). These data, combined with the limited number of peaks and relative lack of Runx motif enrichment in the Runx1KO cells, confirm the specificity of the antibody used for the ChIP assay to identify genomic regions are bound by Runx1. The second most enriched class of motifs were for the AP-1 family, a heterodimer composed of proteins belonging to the c-Fos, c-Jun families and shown previously to be co-enriched with Runx1 [26] and mark enhancers in most cell types (27). This suggests a role for Runx1 in maintaining enhancer accessibility to allow binding by other cell-type specific TFs [27–30].

To further determine whether these Runx1 bound regions were involved in transcriptional regulation, we assigned genes to the Runx1 ChIP peaks using the GREAT annotation tool [31] and compared the genes near immunoprecipitated chromatin to the genes that exhibiting expression changes of over 2-fold in Runx1KO cells (S1 Table). This analysis showed that 49% (514/1041) of downregulated genes in Runx1KO cells had a Runx1 ChIP-seq peak in their vicinity, a 1.87 fold enrichment over what was expected by chance (hypergeometric p-value $1.71 \times 10^{-58}$). In contrast, only 34% (545/1599) of upregulated genes had a nearby immunoprecipitated peak (a 1.41 fold enrichment). While the limited overlap of Runx1 ChIP peaks near upregulated genes cannot exclude the possibility that some of these may represent sites of Runx1 mediated repression, the results are consistent with the loss of expression in Runx1KO cells of direct Runx1 targets and less consistent with a model in which upregulated genes were repressed directly by Runx1.

The failure of Runx2 to regulate the same genes as Runx1, as reflected in the RNA-seq data (Fig 2A), might reflect differential occupancy of certain genomic regions by Runx1, and not Runx2, due to differences in DNA binding preferences or protein interaction partners. Alternatively, Runx1 and Runx2 might occupy the same loci, but Runx2 might have different effects on gene expression compared to Runx1. To further investigate these possibilities, we performed Runx2 ChIP-seq in both control and Runx1KO cells. The Runx2 ChIP identified combined sets of 17,595 peaks present in control cells and 15,608 peaks present in Runx1KO cells, with the Runx motif strongly enriched in both cell types (p-value $1 \times 10^{-2244}$ and $1 \times 10^{-2363}$, respectively (S3 Table)). Comparisons between the chromatin bound by Runx1 and Runx2 revealed remarkable overlap of the peaks in control cells, and retention of Runx2 ChIP signal at Runx1 peaks in Runx1KO cells (Figs 2B, 2D, and S2A). The majority of Runx1 peaks are enriched for Runx2 ChIP reads relative to adjacent background regions. Regulatory Element Locus Intersection (RELI) analyses [32] confirmed the highly significant agreement between

the Runx1 (mk4), Runx2 (mk4), and Runx2 (Runx1 KO) ChIP-seq datasets (control cell 194.46 fold enriched, p-value 2.0 x $10^{-219}$; Runx1KO cell 177.85 fold enriched, p-value 2.32 x $10^{-219}$; S2C Fig and S4 Table). These results suggest that the regulatory regions near downregulated genes in Runx1KO cells retain Runx2 binding, which evidently is not sufficient to drive their expression. Notably, enrichment analysis revealed that the Runx1 ChIP regions overlapped very significantly with FOSL1 and various Jun proteins in other cell types, consistent with the observed AP-1 motif enrichment in our ChIP datasets and supporting the hypothesis that Runx1 binds to enhancers. Accordingly, the Runx1 ChIP peaks enriched significantly for enhancer marks such as H3K27 acetylation, H3K4 methylation, H3K4 dimethylation, H3K4 trimethylation, and DNase sensitivity in multiple cell types, strengthening the assertion that the Runx1 bound regions are likely enhancers. We confirmed these associations by performing H3K27 acetylation ChIP-seq in control and Runx1KO cells, which revealed that Runx1 ChIP-seq peaks significantly overlap with H3K27 acetylation peaks that were stronger in control cells compared to Runx1KO cells (33.36 fold, p-value 2.17 x 10–182) (S4 Table). Filtering Runx1 ChIP peaks only to those that overlapped with predicted AP-1 binding showed similar enrichment for H3K27 acetylation in control cells (Runx1 enrichment 35.48 fold, p-value 2.83 x 10–180). By contrast, Runx1 ChIP peaks that did not overlap with predicted AP-1 binding sites were not significantly enriched for control cell specific H3K27acetylation. Overall, these analyses indicate that Runx1 has transcriptional activator function at enhancers in a large subset of its target loci. Runx2 can also bind these enhancers but is not sufficient to maintain open chromatin or to activate the expression of the associated genes.

## Dramatic changes in chromatin accessibility drive expression changes in Runx1KO cells

The Runx2 ChIP data indicated that most regulatory regions remained accessible to Runx2 binding in Runx1KO cells (Fig 2B). To examine chromatin accessibility near down regulated genes in Runx1KO cells, we next performed ATAC-seq experiments. As with the RNA-seq and ChIP-seq data, all ATAC-seq replicates were highly reproducible (S3A and S3B Fig). The majority of open chromatin regions represented by 37,481 ATAC-seq peaks displayed similar levels of reads between control and Runx1KO cells, henceforth called Runx1-independent ATAC-seq peaks (Fig 3A, intersect in Venn diagram, and heatmap in Fig 3B, left panel). Notably, we also observed substantial and reproducible loss in chromatin accessibility after Runx1 was deleted– 8,741 genomic loci had significantly lower accessibility in Runx1KO cells vs control, which we denote as Runx1-dependent peaks (Fig 3A, left unique area in Venn diagram and heatmap in Fig 3B, middle panel). Interestingly, a similar number of regions (9,427) showed increased accessibility in Runx1KO cells vs control (Runx1KO-induced; Fig 3A right unique area in Venn diagram and heatmap in Fig 3B, right panel). We denote these sites as Runx1KO-induced.

These changes in chromatin accessibility could reflect Runx1 functioning as an activator at some loci (i.e., by opening or maintaining the accessibility of Runx1-dependent sites) and a repressor at others (i.e., by keeping Runx1KO-induced sites inaccessible). If Runx1 is directly acting as both an activator and a repressor in this manner, then both classes would be expected to be enriched for Runx1 motifs. To test this hypothesis, we repeated the analyses described above and again found strong enrichment for AP-1 motifs (p-value 1 x $10^{-1601}$), further implicating Runx1-dependent ATAC-seq regions as enhancers (Fig 3C and S3 Table). The 2nd most enriched motif class in Runx1-dependent open chromatin regions was Runx (p-value 1 x $10^{-388}$) and as expected, the dataset had highly significant overlap with our Runx1 ChIP data (26.43 fold enriched, Bonferroni corrected p-value 2.16 x $10^{-211}$; S4 Table). This enrichment of

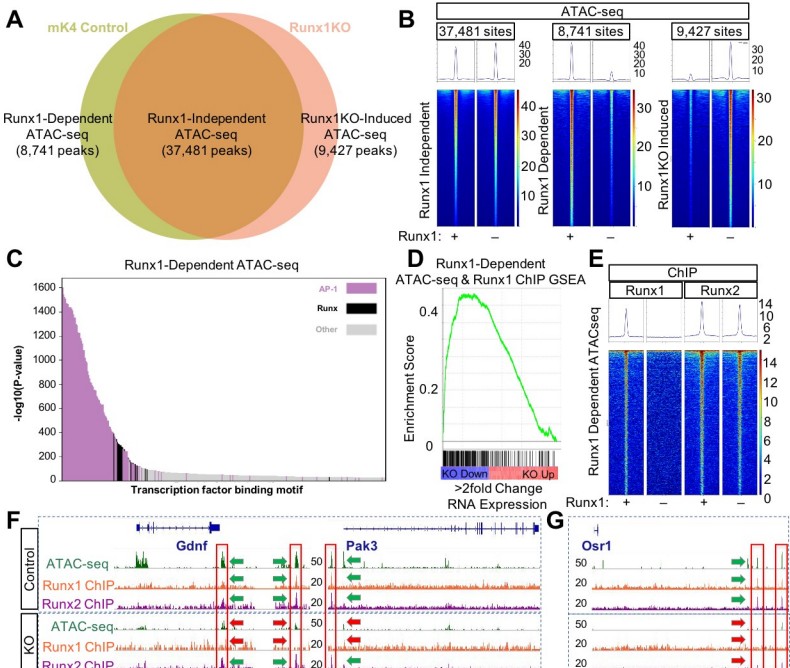

**Fig 3. Runx1 Deletion Alters Chromatin Accessibility Despite the Presence of Runx2.** A) Venn diagram of ATAC-seq peaks in control and Runx1KO cells showing the number of regions open in both cell lines (Runx1-independent), regions open only in control cells (Runx1-dependent) and regions open only in Runx1KO cells (Runx1KO-induced). B) Heatmaps of the ATAC-seq reads mapping to Runx1-independent, Runx1-dependent, and Runx1KO-induced peaks in the control and Runx1KO cells. C) Graph of the -log10 p-values of motif enrichment, displaying that Runx1-dependent ATAC-seq peaks are strongly enriched for AP-1 and Runx motifs. D) Gene set enrichment analysis showing enrichment of transcriptionally down-regulated genes (in blue and up-regulated in red) by Runx1-dependent ATAC-seq peaks that are bound by Runx1. E) Heatmap showing Runx1-Dependent ATAC-seq regions bound in Runx1 and Runx2 ChIP experiments. F) Genomic snapshots of *Gdnf* and *Pak3* genes that are downregulated in Runx1KO cells showing open chromatin regions present in control cells but not Runx1KO cells that are bound by Runx1 and Runx2 and retain Runx2 binding in the Runx1KO cells. G) Genomic snapshot of the Runx1KO downregulated gene *Osr1* that has genomic regions that lose chromatin accessibility in Runx1KO cells, which are bound by Runx1 and Runx2 in control cells, with reduced Runx2 binding in Runx1KO cells.

Runx1 ChIP peaks within Runx1-dependent ATAC-seq peaks was nearly 10 fold higher than the enrichment of Runx1 ChIP peaks within Runx1KO-induced ATAC-seq peaks (5.2 fold enrichment, corrected p-value 9.1 x $10^{-67}$), consistent with a role for Runx1 in promoting and maintaining open chromatin in these cells, but not with Runx1 maintaining repression. Since enrichment analysis found significant enrichment for our mK4 Runx1 ChIP sites in AML (10.73 fold enriched, p-value 8.42 x $10^{-168}$) and HPC-7 cells (9.73 fold enriched, p-value 1.13 x $10^{-143}$; S4 Table), these may be functionally important Runx1-dependent enhancers in multiple cellular contexts. We next assigned Runx1-dependent ATAC-seq regions to nearby Runx1-dependent transcripts using the GREAT annotation tool [31]. Runx1-dependent ATAC-seq regions were enriched 3.12 fold near down-regulated genes in Runx1KO cells (hypergeometric p-value = 4.80 x $10^{-200}$), as expected from Runx1-dependent maintenance of enhancers regulating expression of nearby genes.

Runx1 ChIP peaks obtained in control cells revealed extensive Runx1 binding within both Runx1-dependent peaks and peaks that remain open in the absence of Runx1 (Figs 3A and 3B and S3C Fig). However, regions that are bound in the Runx1 ChIP but become inaccessible in Runx1KO cells (Runx1-dependent) show strong enrichment (4.08 fold, hypergeometric p-value 1.74 X $10^{-67}$) for proximal genes whose expression decreases in Runx1KO cells

(Fig 3D). Thus, the combination of ATAC-seq and ChIP data helps to define a set of functional, Runx1-dependent regulatory regions in the mK4 genome and supports the conclusion that Runx1 activator function is related to its role in maintaining accessible chromatin in mK4 cells.

The Runx1-dependent regions that become less accessible in the absence of Runx1 in Runx2-expressing cells might do so because those specific regions are bound only by Runx1 and not by Runx2. To test this hypothesis, we compared the Runx2 ChIP-seq reads with the three classes of ATAC-seq peaks shown in Fig 3A. Notably, the Runx2 ChIP signal was present at both Runx1-independent and Runx1-dependent ATAC-seq regions (Fig 3E), and in both mK4 and Runx1KO cells (S3B Fig). For example, we show that putative enhancers located near the Runx1 regulated genes *Gdnf* and *Pak3* [33–37] (Fig 3F, green arrows) became inaccessible In Runx1KO cells (Fig 3F, red arrows) while still retaining Runx2 binding (Fig 3F, green arrow). Further examples are provided as supplementary data to demonstrate that this pattern of retained Runx2 binding in the absence of ATAC-seq peak typifies genes whose expression is reduced in Runx1KO cells (S4A Fig; reproducibility between triplicates shown in S5A and S5B Fig). These data suggest that while Runx2 can bind to closed chromatin like Runx1 [20], it cannot make the chromatin accessible to other factors.

Other Runx1-dependent regions display greatly reduced Runx2 binding. For example, two potential enhancers near *Osr1*, a reported Runx target gene [38], are open and bound by both Runx1 and Runx2 in control cells (Fig 3G, green arrows), but become inaccessible with limited binding of Runx2 in the Runx1KO cells (Fig 3G, red arrows). To ask if sites that lose Runx2 binding in Runx1KO cells were enriched near downregulated genes, we separated the Runx1-dependent ATAC-seq regions bound by Runx1 into two groups: those sites that had an overlapping Runx2 ChIP peak in Runx1KO cells and those sites that did not (S4B Fig). Enrichment analysis on these two groups, performed as above, revealed similar strong enrichment for downregulated genes near the sites immunoprecipitated by Runx2 in Runx1KO cells (4.14 fold enrichment, hypergeometric p-value $5.41 \times 10^{-60}$) and near sites not immunoprecipitated by Runx2 in Runx1KO cells (4.12 fold enrichment, hypergeometric p-value $9.01 \times 10^{-15}$). Collectively, the Runx1 and Runx2 ChIP data and subsequent enrichment analyses indicate that Runx1 binds to and promotes or maintains chromatin accessibility at a large number of loci, many of which are associated with Runx1-responsive genes. Despite remaining bound to these same regions in the absence of Runx1, Runx2 is unable to compensate for the lack of Runx1.

## Runx1KO-induced chromatin regions are opened due to the loss of Zeb transcriptional repressors

Our analyses above suggest that while Runx1KO-induced ATAC-seq regions require Runx1 to remain inaccessible, the lack of Runx1 binding to these regions is consistent with indirect repression by Runx1. To explore the possibility that particular Runx1-dependent protein(s) are maintaining repression, we performed TF binding motif enrichment analysis at these sites. Indeed, Runx motifs were absent from the top 1,500 enriched motifs (S3 Table), consistent with an indirect mechanism whereby Runx1 acts either by repressing an activator or by activating a repressor protein. Motif enrichment analysis of Runx1KO-induced ATAC-seq peaks compared to Runx1-dependent ATAC-seq sites revealed significant enrichment for the Zeb repressor motif (Figs 4A and S6A), consistent with a mechanism in which Runx1 regulates Zeb expression which in turn actively maintains inaccessible chromatin architecture at multiple sites. Accordingly, we observed that the levels of Zeb1 and Zeb2 mRNA were over 10-fold lower in Runx1KO cells (Fig 4B), confirmed independently by real-time quantitative PCR (RT-qPCR; S6D Fig). Using Western blot analysis, we found that the Zeb1 protein is expressed

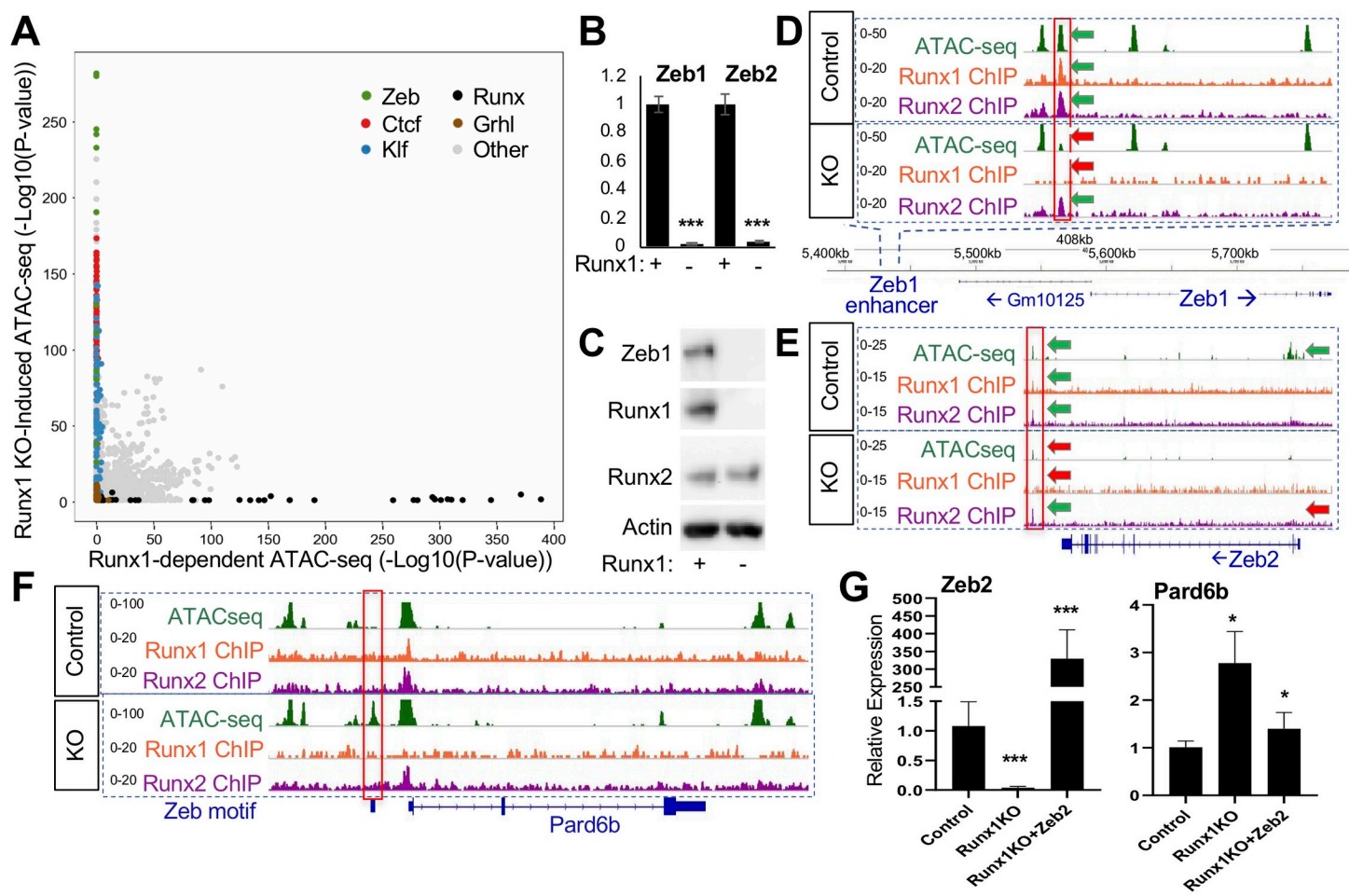

**Fig 4. Runx1KO Cells Lack Zeb Repressors, Leading to the Opening of Chromatin.** A) Graph displaying p-values of transcription factor motif enrichment in Runx1-dependent versus Runx1-induced ATAC-seq, revealing that Zeb motifs are specifically enriched in Runx1KO-Induced ATAC-seq peaks. Additionally, Ctcf, Klf, and Grhl motifs are enriched in the Runx1KO-induced ATAC-seq peaks, while Runx motifs are enriched in the Runx1-dependent ATAC-seq. Note that this graph has had AP-1 motif enrichment results removed in order to focus on other motif enrichment levels (see S6A Fig for all transcription factor motifs). B) RT-qPCR showing that Runx1KO cells lose expression of Zeb1 and Zeb2. C) Western blot showing the absence of the Zeb1 protein in Runx1KO cells. D) Genomic snapshot showing a chromatin region near *Zeb1* that is bound by Runx1 and Runx2 and loses chromatin accessibility in Runx1KO cells. E) Genomic snapshot of the *Zeb2* locus showing a downstream potential enhancer bound by Runx1 and Runx2 that has decreased chromatin accessibility along with a loss of expression in Runx1KO cells. F) Genomic snapshot of the Zeb target gene *Pard6b* locus showing a promoter region containing a predicted Zeb binding site that is specifically open in Runx1KO cells. G) RT-qPCR showing that Zeb2 transient transfection of Runx1KO cells induces repression of *Pard6b* expression down to levels similar to those in control cells after 1 day of selection for transfected cells followed by 2 days of growth in media. (* = p-value < 0.05, ** = p < 0.005; *** = p-value < 0.0005).

in control cells but not in Runx1KO cells (Fig 4C). Unfortunately, commercially available antibodies to Zeb2 failed to detect it in control cells. Next, we asked if Runx1 was a direct regulator of Zeb1 expression by examining the Zeb1 locus for Runx1 binding. An accessible regulatory region near Zeb1 was bound by Runx1 and Runx2 in control cells (Fig 4D green arrows), but became inaccessible in Runx1KO cells (Fig 4D, red arrows). Similarly, an accessible regulatory region near Zeb2, bound by Runx1 and Runx2 in control cells, became inaccessible in Runx1KO cells (Fig 4E). Additionally, the Zeb1 and Zeb2 TSS, open in control cells, were less accessible in Runx1KO cells (Fig 4E and S6B), mirroring the dramatic differences in the expression levels of Zeb1 and Zeb2 between control and Runx1KO cells. These data suggest that both Zeb1 and Zeb2 are regulated by Runx1-dependent enhancers in mK4 cells.

In support of the hypothesis that the opening of the chromatin in Runx1KO cells is due to the loss of Zeb repressors, we examined a known Zeb1 repressed target, *Pard6b*, that was

upregulated in Runx1KO cells nearly 10-fold based on RNA-seq analysis (S6C Fig) and RT-qPCR (S6D Fig). A predicted Zeb binding site upstream of *Pard6b* is only accessible in Runx1KO cells (Fig 4F). Thus, we hypothesized that a subset of the genes upregulated in the Runx1KO cells may be the result of losing Runx1-dependent Zeb 1 and Zeb2 expression, and the subsequent derepression of Zeb targets.

To test whether Zeb expression was sufficient abolish gains in gene expression seen in Runx1KO cells, we transiently transfected Runx1KO cells with an expression vector driving Zeb2 mRNA levels 330 fold over baseline as determined by RT-qPCR (Fig 4G, left—recall that there is no usable antibody to Zeb2). Next, we tested the expression of *Pard6b* in Zeb2-transfected Runx1KO cells and found that its expression level was significantly reduced (Fig 4G, right). Because *Pard6b* expression levels did not fully return to control levels, there may be additional proteins contributing to its regulation. Alternatively, stable expression of Zeb might be required over a longer period. This supports the hypothesis that many of the upregulated genes in Runx1KO cells may be indirectly affected through the loss of Zeb-mediated repression.

Not all of the open chromatin gained in Runx1KO cells contains Zeb sites. Grhl motifs are significantly enriched in the small subset of Runx1KO-induced ATAC-seq fragments that lack Zeb motifs (Fig 4A and S3 Table). A genomic snapshot of the *Grhl2* loci reveal Zeb motif-containing chromatin regions that become accessible in Runx1KO cells (Fig 5F), concomitant with increased *Grhl2* expression, suggesting that Grhl2 was suppressed by Zeb in mK4 cells. As we had observed for *Pard6b*, *Zeb2*-expressing Runx1KO cells display significant downregulation *Grhl2* expression by RT-qPCR (Fig 5A), consistent with the hypothesis that Zeb proteins are sufficient to restore repression.

To directly test whether loss of Zeb expression is responsible for the opening of chromatin and gene upregulation observed in Runx1KO cells, we again utilized CRISPR-Cas9 to knock-out both Zeb1 and Zeb2 to generate ZebDKO cells. We successfully obtained several clones that deleted the exons containing the start codon for each (Fig 5B) and confirmed the loss of Zeb1 protein (Fig 5C). Global expression analysis by RNA-seq confirmed that much of the transcriptional deregulation observed in Runx1KO cells was recapitulated in ZebDKO cells, as shown in a scatter plot comparing the differentially expressed genes (DEG) in Runx1KO vs. control mK4 cells with those in ZebDKO vs. control mK4 cells (Fig 5E). These results are further illustrated in a heatmap of the genes displaying over 2 fold change in expression levels in the replicates comparing the control cells to the Runx1KO and ZebDKO cells (S7C Fig). Gene Set Enrichment Analysis (GSEA) showed enrichment of genes upregulated in Runx1KO cells among genes significantly increased in ZebDKO cells (S7D Fig). Specifically, we confirmed that *Grhl2*, *Pard6b* and *Itgb4* were all upregulated in Runx1-expressing ZebDKO cells by RT-qPCR (Figs 5D and S7A and S7B). The enrichment of Grhl motifs in the Runx1KO-induced ATAC-seq and the upregulation of Grhl2 targets (Ovol1, Cldn4, and Cgn) in both Runx1KO and ZebDKO RNA-seq datasets (S1 Table) suggest that *Grhl2* derepression has important functional consequences *in vitro*. Moreover, *Grhl2* repression by Zeb plays a critical role in reciprocal negative feedback loops between these factors during epithelial-mesenchymal transition (EMT) [39]. These observations prompted us to analyze the regulatory region of *Grlh2* as a representative Zeb target. To determine whether the transcriptional upregulation in ZebDKO cells was due to opening of chromatin near this DEG, we performed ATAC in ZebDKO cells and used QPCR to compare the enrichment of several putative enhancers near the *Grhl2* locus (-40kb, -69kb, and -74kb) near the Grhl2 locus to their status in the control mK4 cells. The accessible areas gained in Runx1KO cells were also more accessible in the ZebDKO cells relative to control. By contrast, constitutively open (-32kb) or inaccessible (-36kb) regions were indistinguishable between the control and ZebDKO cells (Fig 5G). Combined, Zeb knockout in Runx1-expressing cells and overexpression in Runx1KO cells

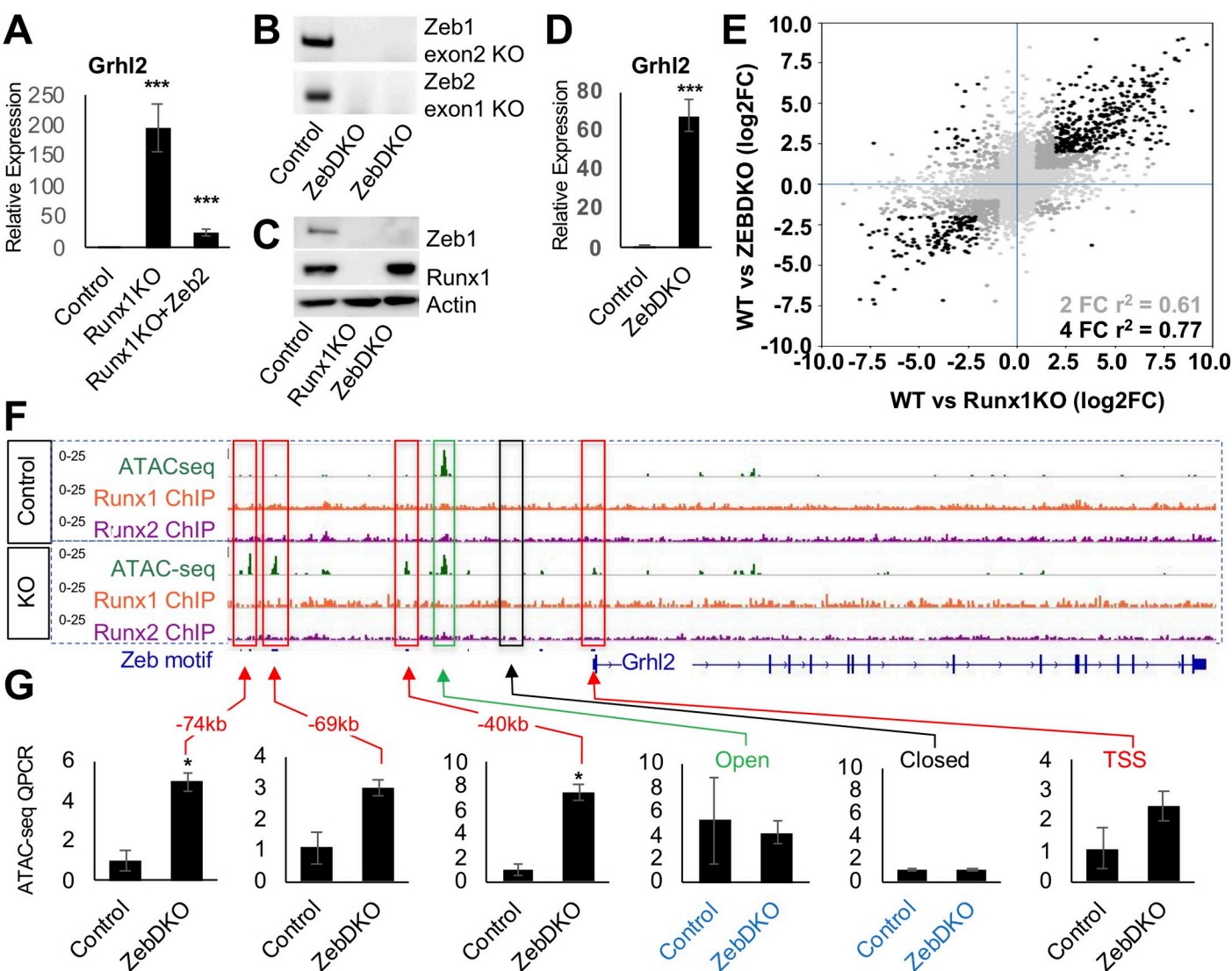

**Fig 5. Loss of Zeb Proteins Recapitulates the Opening of Chromatin and Transcriptional Induction of Runx1KO.** A) RT-qPCR confirmation of the upregulation of *Grhl2* in Runx1KO cells, which is suppressed by transient transfection of Zeb2. B) Genotyping PCR showing deletions of Zeb1 exon2 and Zeb2 exon1 in ZebDKO clones. C) Western blot confirmation of loss of Zeb1 proteins in ZebDKO cells. D) RT-qPCR showing that ZebDKO cells have a dramatic increase in Grhl2 expression. E) Scatter plot showing the correlation of gene expression changes in Runx1KO and ZebDKO cells compared to control mK4 cells (light gray all genes, dark gray genes with 2 fold change in expression in both and black are genes with over 4 fold change in expression in both). F) *Grhl2* loci displaying ATAC-seq regions that are specifically open in Runx1KO cells (outlined in red) that contain predicted Zeb binding sites. G) Graphs of ATAC QPCR data showing that the regions near *Grhl2* that were opened in Runx1KO cells similarly become significantly more open in ZebDKO cells while control open and closed regions are similar between control and ZebDKO cells. (* = p-value < 0.05, ** = p < 0.005; *** = p-value < 0.0005).

demonstrate that Zeb proteins are both required and sufficient to suppress a cohort of targets upregulated in Runx1KO cells, and the loss of Runx1 unleashed a TF cascade due to the dere-pression of Zeb. Thus, these results reveal Runx1 to be key factor whose loss creates ripple effects perturbing the entire transcriptional network (Fig 6).

## Discussion

We report herein that Runx1 regulates the chromatin landscape directly and indirectly to broadly impact transcription at multiple loci in a mouse kidney-derived cell line. Even though

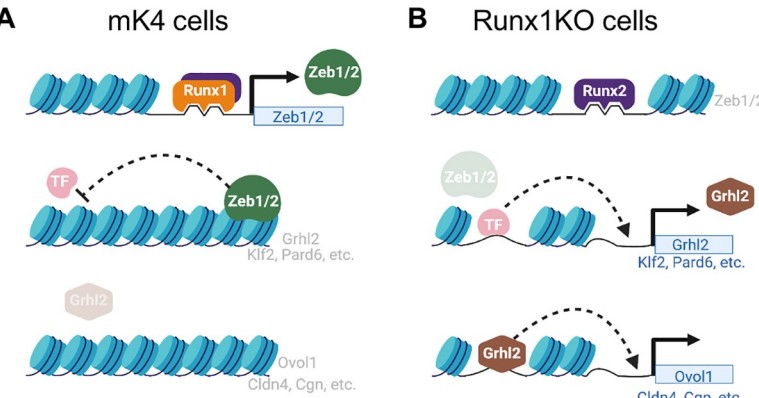

**Fig 6. Model of Transcription Factor Network Perturbation by Runx1-Deficiency.** A) In control mK4 cells, Runx1 induces the transcriptional repressors Zeb1 and Zeb2 that inhibit other transcriptional activators such as Grhl2, resulting in inhibition of downstream Grhl2 target genes. B) Runx1KO cells lose expression of Zeb1 and Zeb2, which derepresses their targets including Grhl2 and Klf2, which in turn leads to upregulation of their downstream targets such as Ovol1, Cldn4, and Cgn. Figure was created using BioRender.com.

Runx1 has been reported to be both a transcriptional activator and a repressor [4,40,41], our analyses suggest that Runx1 functions primarily as an activator in mK4 cells despite the presence of its co-repressor Gro/TLE1-3 (S1 Table). Runx1 does so by maintaining chromatin accessibility directly at many loci, including at the Zeb1 and Zeb2 loci. The Zeb proteins in turn maintain inaccessible chromatin at many loci, including those encoding several other transcription factors (e.g., Grhl2), cascading into further downstream TF targets *Grlh2* regulates (e.g., Ovol1). Runx1 deficiency leads to loss of Zeb protein expression and subsequently to gained accessibility and expression of Zeb-repressed genes, including additional transcriptional activators such as *Grhl2*, *Ovol1* and *Klf2* (Fig 6), leading ultimately to widespread genome-wide changes to chromatin accessibility and transcription.

Interestingly, these broad effects on chromatin accessibility and transcription occur despite the presence of Runx2 in these cells. Runx2 remains bound to most of the sites immunoprecipitated by Runx1, including inaccessible chromatin, but is unable to compensate for Runx1 loss and maintain accessibility of chromatin [20]. The molecular mechanism underlying the specific ability of Runx1 to maintain chromatin accessibility near its target genes remains under investigation.

In addition to facilitating transcription by enhancing accessibility, TFs can regulate transcription by recruiting additional factors or by contributing activity to preassembled complexes. In mK4 cells, Runx1 appears to largely act by maintaining chromatin accessible to other TFs, and by maintaining the expression of Zeb repressors to prevent a myriad of other sites from responding to their regulators. The enrichment for down-regulated genes near Runx1-dependent ATAC-seq is greater than that observed for Runx1 ChIP, which suggests that impact on accessibility spreads across chromatin regions beyond the sites directly bound by Runx1. This raises the possibility that Runx1 may play an additional role in facilitating the loading of other TFs through the opening or maintaining of enhancer accessibility, to allow full transcriptional activation of its target genes and the integration with other signaling pathways, as has been described for other transcription factors [42,43]. It will be interesting to determine whether Runx1 plays a generalized master-regulator role controlling the activity of other signaling pathways through the modulation of chromatin accessibility.

The role of Runx1 in kidney development and disease has not been extensively investigated. The mK4 cells used in this study are derived from the metanephric mesenchyme, which contains all the nephron progenitors that form the mammalian kidney via mesenchyme-to-

epithelial transition. Runx1 regulates EMT in other tissues [44–47]. While the role of Runx1 during kidney development is unknown, it is activated in kidney during injury and repair where it may contribute to several disease processes [23,48–51], including cancer [52,53], in which it may collaborate with Runx2 [54] to drive EMT associated with worse outcomes. The profound consequences of Runx1 loss to the mK4 transcriptome suggests that a further delineation of its role in the kidney may be warranted.

The finding that gains in chromatin accessibility in Runx1KO cells largely reflects a loss of Zeb repressor protein activity suggests that Runx1 "repressor" functions may be executed by Zeb in many cells and tissues, where Zeb expression requires Runx1. This may reflect an underappreciated and widespread collaboration between these proteins. Accordingly, examination of public functional genomics data revealed that the Runx1-bound Zeb1 enhancer we identified in mK4 cells is accessible (i.e., DNase hypersensitive) in other tissues including several hemopoietic cell lines (S4 Table) and is also bound by Runx1 in AML. Further, gaining Zeb expression could be relevant not only to the role of Runx1 in AML but might also contribute to solid organ malignancies, where Zeb proteins are critical inducers of EMT. In agreement with this hypothesis, co-expression of both Runx1 and Zeb2 in circulating tumor cells has been shown to significantly correlate with cancer reoccurrence [55].

Collectively, our data indicate that loss of Runx1 produces widespread genomic and transcriptional changes through a cascade of direct and indirect sequalae involving Zeb upstream of multiple transcriptional repressors and activators, and reveal key members of this complex network of interacting TFs. Mediating repression indirectly through the upregulation of repressors is widespread (e.g., Hes/Hey upregulation by Notch), and is likely to include additional repressors discoverable by motif enrichment analysis of vanishing ATAC-seq peaks in different genetic models and cellular contexts.

## Materials and methods

### Tissue culture

The mK4 and Runx1KO cells were grown in DMEM supplemented with 10% FBS, L-glutamine, penicillin/streptomycin, and sodium pyruvate. The cells were transfected using Lipofectamine 2000 (Invitrogen) following the manufacturer's directions.

### Generation of Runx1KO or ZebDKO cells

e used the mK4 cell line as our control cell line, as described in [21]. From the control cell line, we generated sub-lines that do not express Runx1 using guide RNAs (gRNAs) in the px458 and px459 CRISPR/Cas9 vectors to delete the third exon of Runx1 containing the start codon utilized in mK4 cells. The ZebDKO cells were generated similarly using gRNAs targeting the second exon of Zeb1 and first exon of Zeb2 that contain the start codons of these genes. The targeting sequences were generated with the method and tools described in [56]. These cells were than transfected with Lipofectamine 2000 (Invitrogen) according to the manufacturer's instructions. The cells underwent selection with 1 μg/ml puromycin for two days. Subsequent clones were picked approximately a week later using cloning disks, expanded, and screened for exon 3 deletion by PCR and for loss of Runx1 protein expression by Western blot. The ZebDKO clones were screened by PCR to the targeted exons and Western blot to Zeb1 for protein expression.

### Western blot

Confluent control cells or Runx1KO mK4 cells were collected in 100 ul of RIPA-DOC with protease inhibitors plus 100 ul of 2X sample buffer. Protein samples were run on 7%

polyacrylamide gels and then transferred to PVDF. The primary antibodies used were as follows: Runx1 (Cell Signaling #8529), Runx2 (Cell Signaling #8486), Runx3 (Cell Signaling #9647), Zeb1 (Bethyl #a301-922a) and Beta-actin (Sigma #A5441). The indicated antibodies were applied at 1:1000 dilutions (except for beta-actin that was used at 1:5000) in 5% milk in PBST overnight at 4 degrees and following 3x wash in PBST the secondary antibodies were applied at 1:5000 dilution in 5% milk in PBST at room temperature for 1hr. The Western blot signal was detected using Thermofisher Supersignal Femto ECL reagent according the manufacturer protocol and imaged using a Bio Rad Chemidoc MP Imaging System.

## RNA-seq

mK4 control, Runx1KO, and ZebDKO cells were cultured in triplicate in standard mK4 media (DMEM plus 10% FBS, 2% L-glutamine, 1% Pen/Strep, and 1% Sodium Pyruvate) on 12 well plates until nearly confluent. Cells were removed from the plate with trypsin that was subsequently inactivated using mK4 conditioned media to prevent a feeding effect from fresh media activating signaling pathways. RNA was collected using Invitrogen's Purelink RNA Mini kit according to the manufacturer's directions. RNA-seq on polyA isolated RNA was performed by the CCHMC sequencing core to produce over 20 million reads per sample.

## RT-qPCR

Biological triplicate samples of RNA were converted to cDNA using Superscript II Reverse Transcriptase from Invitrogen following the manufacturer's protocol. The cDNA was diluted to 40 ng/μl, and 5μl of each sample was added to each RT-qPCR reaction that were amplified using iTaq Universal SYBR Green Supermix from Bio-Rad and read on a StepOnePlus Real-Time PCR System from Applied Biosystems. Gene expression levels were normalized to Gapdh and changes were determined relative to control cells, with significance calculated using Student's t-test.

## ATAC-seq

ATAC-seq experiments were carried out in the control mK4 cells and Runx1KO cells in triplicate. Experiments were performed following the protocol laid out by the Kaestner Lab [57]. The Tn5 used in the experiment was prepared using the method outlined in [58]. The purification of the library prep was done in accordance with [59].

## ChIP-seq

Control and Runx1KO mK4 cells were grown on 10cm plates in triplicate until nearly confluent and removed from the plate using trypsin that was inactivated with conditioned media. Individual cells were counted and $10^6$ cells were used to make the ChIP lysates. Cells were incubated in crosslinking solution (1% formaldehyde, 5 mM HEPES [pH 8.0], 10 mM sodium chloride, 0.1 mM EDTA, and 0.05 mM EGTA in RPMI culture medium with 10% FBS) and placed on a tube rotator at room temperature for 10 min. To stop the crosslinking, glycine was added to a final concentration of 0.125 M and tubes were placed back on the rotator at room temperature for 5 min. Cells were washed twice with ice-cold PBS, resuspended in lysis buffer 1 (50 mM HEPES [pH 8.0], 140 mM NaCl, 1 mM EDTA, 10% glycerol, 0.25% Triton X-100, and 0.5% NP-40), and placed on a tube rotator at 4C for 10 minutes. Nuclei were harvested after centrifugation at 10,000g for 5 min, resuspended in lysis buffer 2 (10 mM Tris-HCl [pH 8.0], 1 mM EDTA, 200 mM NaCl, and 0.5 mM EGTA), and placed on a tube rotator at room temperature for 10 min. Nuclei were collected again by centrifugation at 10,000g for 5 minutes.

Protease and phosphatase inhibitors were added to both lysis buffers. Nuclei were then resuspended in the sonication buffer (10 mM Tris [pH 8.0], 1 mM EDTA, and 0.1% SDS). A S220 focused ultrasonicator (COVARIS) was used to shear chromatin (150- to 500-bp fragments) with 10% duty cycle, 175 peak power, and 200 bursts per cycle for 7 min. A portion of the sonicated chromatin was run on an agarose gel to verify fragment sizes. Sheared chromatin was precleared with 20 μl Dynabeads Protein A (Life Technologies) at 4˚C for 1 hr.

Immunoprecipitation of Runx-chromatin complexes was performed with an SX-8X IP-STAR compact automated system (Diagenode). Beads conjugated to antibodies against Runx1 (Rabbit mAb #8529, Cell Signaling) or Runx2 (Rabbit mAb #8486, Cell Signaling) were incubated with precleared chromatin at 4˚C for 8 hours. The beads were then washed sequentially with wash buffer 1 (50 mM Tris-HCl [pH 7.5], 150 mM NaCl, 1 mM EDTA, 0.1% SDS, 0.1% NaDOC, and 1% Triton X-100), wash buffer 2 (50 mM Tris-HCl [pH 7.5], 400 mM NaCl, 1 mM EDTA, 0.1% SDS, 0.1% NaDOC, and 1% Triton X-100), wash buffer 3 (2 mM EDTA, 50 mM Tris-HCl [pH 7.5] and 0.2% Sarkosyl Sodium Salt), and wash buffer 4 (10 mM Tris-HCl [pH 7.5], 1 mM EDTA, and 0.2% Triton X-100). Finally, the beads were resuspended in 10 mM Tris-HCl (pH 7.5) and used to prepare libraries via ChIPmentation [60].

## Processing of functional genomics data

RNA-seq, ATAC-seq, and ChIP-seq reads (in FASTQ format) were first subjected to quality control using FastQC (v0.11.7) (parameter settings:—extract -o output_fastqc -f R1.fastq.gz) [61]. Adapter sequences were removed using Trim Galore (v0.4.2) (parameter settings: -o folder—path_to_cutadapt cutadapt—paired R1.fastq.gz R2.fastq.gz) [62], a wrapper script that runs cutadapt (v1.9.1) [63] to remove adapter sequences from the reads. The quality-controlled reads were aligned to the reference mouse genome version NCBI37/mm9 using STAR v2.6.1e [64]. Duplicate reads were removed using the program sambamba v0.6.8 (parameter settings: -q markdup -r -t 8 trimmed.bam trimmed_dedup.bam) [65]. Gene annotations for RNA-seq analysis were downloaded from the UCSC Table Browser [66] for the NCBI37/mm9 genome in GTF format. Unnormalized gene expression was assessed by counting features for each gene defined in the NCBI's RefSeq database [67]. Read counting was done with the program featureCounts v1.6.2 from the Rsubread package [68]. Differential gene expression between groups of samples was assessed with the R package DESeq2 v1.26.0 [69]. Differentially expressed genes were defined using a cutoff of 0.3 (30% more expression) and an FDR cutoff of 0.05. Subsequent comparisons to ChIP-seq and ATAC-seq peaks focused on genes displaying over a 2 fold change in expression. For heatmaps, we used the logarithm (base 10) of normalized counts, expressed in transcripts per million (TPM). Plots were created using R base graphics as well as with the ggplot2 package [70].

ATAC-seq and ChIP-seq data were processed using the following steps. Peaks were called using MACS2 v2.1.2 (parameter settings: callpeak -g mm -q 0.01—broad -t trimmed_dedup.bam -f BAM -n trimmed_dedup_peaks) [71]. Specific ChIP peaks were identified by MACS2 peak calling on the combined replicate reads and removing peaks that overlapped with non-specific background peaks called in the Runx1 ChIP in the Runx1KO cells. Peaks shared across experiments (i.e., peaks shared between replicates or shared between treatments/conditions) were identified as peaks with 50% or greater overlap, using BEDtools v2.27.0 [72]. The final peak sets for each condition were obtained by requiring peaks to be present in at least two out of the three biological replicates. When comparing across treatments or conditions, peak overlap between any of the three replicates in either treatment/conditions was considered a shared peak between the treatments/conditions. Final peaks, originally in BED format, were converted to Gene Transfer Format (GTF) format to enable fast counting of reads under the peaks

using the program featureCounts v1.6.2 (Rsubread package) (parameter settings: feature-Counts—ignoreDup -M -t peak -s 0 -O -T 4 -a common_peaks.gtf -o output_counts.txt trimmed_dedup.bam). The resulting matrix of raw counts was normalized for all experiment types to transcripts per million values (TPMs). TF binding site motif enrichment analysis was performed using the HOMER software package [73], which was modified to use a log base 2 scoring system and the set of mouse motifs contained in build 2.0 of the Cis-BP database [25].

Enrichment analysis for functional genomic datasets: We used the RELI algorithm (32) to identify genomic features (TF binding events, histone marks, ATAC-seq peaks, etc.) that significantly overlap with our functional genomics datasets. As input, RELI takes the genomic coordinates of peaks from an input dataset. RELI then systematically intersects these coordinates with each member of a large collection of functional genomics datasets. The number of input regions overlapping the peaks of each dataset is counted. Next, a p-value describing the significance of the overlap of each dataset is estimated using a simulation-based procedure in which the peaks from the input dataset are randomly redistributed within open chromatin regions in mouse cells. (These open chromatin regions were created from the union set of publicly available mouse DNase-seq datasets obtained from mouse ENCODE [74]. A distribution of expected overlap values is created from 2,000 iterations of this randomly sampling process. The distribution of the expected overlap values from the randomized data resembles a normal distribution and can thus be used to generate a Z-score and corresponding p-value estimating the significance of the observed number of input regions that overlap each dataset.

## Supporting information

**S1 Fig. Western blot for Runx1, Runx2, Runx3, and actin in C2C12, mK4 and mK4-Runx1KO cells showing that all three Runx proteins are expressed in C2C12, Runx1 and Runx2 in mK4 cells and only Runx2 in the mK4-Runx1KO cells.**
(PDF)

**S2 Fig.** A) Heatmaps of Runx1 and Runx2 ChIP reads mapped to Runx1 ChIP peaks showing the reproducibility of the ChIP replicates. B) Correlation matrix for Runx1 (left) and Runx2 (right) ChIP showing strong correlation between samples C) Bar graphs of transcription factor motif enrichment -log10 p-values in the Runx1 ChIP in mK4 cells and Runx2 ChIP in mK4 and Runx1KO cells that confirm the strongest enrichment of Runx motifs.
(PDF)

**S3 Fig.** A) Heatmap of Z-scores of individual ATAC-seq samples from mK4 or Runx1KO cells in Runx1-dependent or Runx1KO-induced ATAC-seq peaks showing the reproducibility of the ATAC-seq signal in the replicates and the differences between the mK4 and Runx1KO cells. B) Venn diagrams displaying the overlap of the ATAC-seq peaks between replicates. C) Heatmaps of Runx1 or Runx2 ChIP showing Runx1 binding to both the Runx1-independent and Runx1-dependent ATAC-seq peaks but very little binding to the Runx1KO-induced ATAC-seq peaks. D) Bar graphs of -log10 p-values of motif enrichment in the 3 different classes of ATAC-seq peaks. Runx1-independent ATAC-seq peaks are enriched for AP-1 and Ctcf motifs, Runx1-dependent ATAC-seq enrich for AP-1 and Runx motifs, and Runx1KO-Induced ATAC-seq display enrichment of AP-1, Zeb, and Ctcf motifs.
(PDF)

**S4 Fig.** A) Genomic snapshots of Runx1 and Runx2 ChIP and ATAC-seq around the Runx1KO downregulated genes *Twist2*, *Tnc*, *Zfp810*, *Inhbb*, *Lef1*, and *Igfbp4* that shows Runx1 and Runx2 binding to ATAC-seq peaks that are lost in Runx1KO cells. B) Heatmaps Runx1 or Runx2 ChIP on Runx1-dependent ATAC-seq peaks split into those sites that retain Runx2

binding in Runx1KO cells or sites where Runx2 binding is lost. The Runx1-dependent ATAC-seq that lose Runx2 binding in Runx1KO cells fail to enrich for genes that are down-regulated in Runx1KO cells more than the Runx1-dependent ATAC-seq that retain binding of Runx2.
(PDF)

**S5 Fig.** A) Genomic snapshots of aforementioned Runx1 target genes showing the reproducibility of the signal in all three of the replicates for the ATAC-seq, Runx1-ChIP, and Runx2-ChIP. B) Additional examples not discussed in the text.
(PDF)

**S6 Fig.** A) Plot showing transcription factor motif enrichment (-log10 p-value) in Runx1-dependent versus Runx1KO-induced ATAC-seq peaks that shows AP-1 motifs are strongly enriched in both but Zeb, Ctcf, Klf, and Grainyhead motifs are specific to Runx1KO-induced ATAC-seq peaks while Runx1 motifs are enriched only in Runx1-dependent ATAC-seq peaks. B) Genomic snapshot of Zeb1 that shows the loss of ATAC-seq signal at the TSS that is lost in Runx1KO cells, which is consistent with the loss of expression in these cells. C) Graph of normalized RNA-seq reads showing the upregulation of Pard6b in Runx1KO cells as expected for a Zeb repressed gene. D) RT-qPCR analysis confirming the dramatic down-regulation of both Zeb1 and Zeb2 and upregulation of Pard6b in Runx1KO cells, but lack of difference for the unrelated gene Hes1.
(PDF)

**S7 Fig.** A) RT-qPCR showing that Runx1KO and ZebDKO cells have similar upregulation of *Pard6b*, *Grhl2*, and *Itgb4*. B) Genomic snapshots of *Pard6b*, *Grhl2*, and *Itgb4* displaying chromatin opening of potential enhancers containing Zeb motifs in all three ATAC-seq replicates (red boxes) from Runx1KO cells. C) Heatmap of genes displaying over 2 fold change in expression in Runx1KO and ZebDKO cells compared to control mK4 cells. D) GSEA analysis showing enrichment for genes that were increased over 2 fold in Runx1KO cells in the set of genes that were increased by over 2 fold in the ZebDKO cells, as expected given the known repressive function of Zeb proteins.
(PDF)

**S1 Table. This table is an excel file containing the RNA-seq expression data presented as the transcripts per million (TPM), the genes displaying over 2 fold change in expression with an FDR of less than 0.05 between the Runx1KO and control or ZebDKO and control cells, and the hypergeometric comparison of the gene expression changes with the genes associated with ChIP-seq or ATAC-seq peaks.**
(XLSX)

**S2 Table. This excel file contains the Gene Ontology analysis of the biological pathways that are enriched in the genes that are either upregulated or downregulated by over 2 fold in the Runx1KO cells compared to the control cells.**
(XLSX)

**S3 Table. This table contains the HOMER transcription factor motif enrichment analyses for the various ChIP-seq and ATAC-seq peaks as separate tabs.**
(XLSX)

**S4 Table. This excel file contains the RELI analyses of the ChIP-seq and ATAC-seq peaks that shows the significant overlap with other genomic datasets.**
(XLSX)

**S1 Data. This file contains the primer and gRNA sequences used in this manuscript.**
(XLSX)

**S2 Data. This file has the underlying RT-QPCR data used to the generate the graphs in this manuscript.**
(XLSX)

## Acknowledgments

Dr. Eric Brunskill, Hope Rowden, Carmy Forney, Kevin Ernst, and Dr. Xiaoting Chen for critical discussions, technical assistance, organization, and/or computational help in the production of this manuscript.

## Author Contributions

**Conceptualization:** Matthew R. Hass, Matthew T. Weirauch, Raphael Kopan.

**Data curation:** Matthew R. Hass, Omer Donmez.

**Formal analysis:** Matthew R. Hass, Sreeja Parameswaran, Mario Pujato.

**Funding acquisition:** Matthew T. Weirauch, Raphael Kopan.

**Investigation:** Matthew R. Hass, Daniel Brissette, Omer Donmez.

**Methodology:** Daniel Brissette, Omer Donmez.

**Supervision:** Leah C. Kottyan, Matthew T. Weirauch, Raphael Kopan.

**Visualization:** Sreeja Parameswaran, Mario Pujato.

**Writing – original draft:** Matthew R. Hass.

**Writing – review & editing:** Leah C. Kottyan, Matthew T. Weirauch, Raphael Kopan.

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
