## [Decision Letter · Decision Letter 0]

7 Dec 2020

Dear Dr Kopan,

Thank you very much for submitting your Research Article entitled 'Runx1 Shapes the Chromatin Landscape Via a Cascade of Direct and Indirect Targets' to PLOS Genetics.

The manuscript was fully evaluated at the editorial level and by independent peer reviewers. The reviewers appreciated the attention to an important problem, but raised some substantial concerns about the current manuscript. Based on the reviews, we will not be able to accept this version of the manuscript, but we would be willing to review a much-revised version. We cannot, of course, promise publication at that time.

If you decide to revise the manuscript for further consideration at PLOS Genetics, please aim to resubmit within the next 60 days, unless it will take extra time to address the concerns of the reviewers, in which case we would appreciate an expected resubmission date by email to plosgenetics@plos.org.

[LINK]

We are sorry that we cannot be more positive about your manuscript at this stage. Please do not hesitate to contact us if you have any concerns or questions.

Yours sincerely,

John M. Greally, D.Med., Ph.D.

Section Editor: Epigenetics

PLOS Genetics

Gregory Barsh

Editor-in-Chief

PLOS Genetics

Reviewer's Responses to Questions

**Comments to the Authors:**

Reviewer #1: The manuscript reports the contribution of the Runx1 transcription factor to chromatin organization that is functionally linked to control of gene expression. Utilizing CRISPR-ablation of Runx1, the consequences for chromatin accessibility, evaluated by ATAC seq, was examined and related to gene expression that was assessed by RNA-seq. The observed modifications in gene expression are consistent with well-established parameters of biological control and pathology that have been reported for Runx transcription factors. The influence on the Runx1 mediated genomic/epigenomic landscape was directly addressed by Chip seq. The shared and unique activities of Runx1 and Runx2 were examined with insight into Runx control of proximal and distal genomic regulatory domains. The influence of Runx1 on Zeb1 and Zeb2 expression, within the context of chromatin accessibility was reinforced by overexpression experiments. And, while somewhat un-physiological, these experiments provided insight into regulatory cascades that are linked to Runx regulation of the genomic landscape. The experimental approaches were straightforward and findings were appropriately discussed in relation to results in the literature, particularly those secured from studies on Runx1 control in mammary epithelial and breast cancer cells. The reported findings are a valuable contribution to advancing understanding of Runx1 control of the chromatin landscape and competency for gene expression. The results provide a blueprint for mechanistically exploring Runx regulation of epigenetic control and chromatin organization that supports transcriptional competency.

Reviewer #2: The transcription factor RUNX1 plays important roles in regulating gene expression in many cell types, including hematopoietic, skin, and bone cells. It appears to have positive effects on some genes and repressive effects on other. This manuscript delves more deeply into positive and negative roles of RUNX1 in the mouse kidney cell line mK4, which is derived from metanephric mesenchymal cells. This cell line can be considered an informative model for cells undergoing the mesenchymal to epithelial transition, a process important in development of the kidney. The role of RUNX1 in kidney development has not been extensively studied. The authors generated a RUNX1 knockout (KO) subline of mK4 cells, and determined the impact of RUNX1 loss on the transcriptomes by RNA-seq, the genome-wide binding profiles for RUNX1 and RUNX2 by ChIP-seq, and chromatin accessibility by ATAC-seq. All these determinations were in triplicate to allow stringent determinations of changes. The results show that primary role of RUNX1 in this kidney cell context is that of a transcriptional activator. Most of the genes whose gene expression is positively regulated by RUNX1 (down-regulated in the KO) have, within a reasonable vicinity, plausible candidate enhancers bound by RUNX1. In contrast, most of the genes whose expression is negatively regulated by RUNX1 (up-regulated in the KO) do not appear to be direct targets of RUNX1. Instead, studies of motif enrichment and further biochemical examination reveal a role for ZEB1 and ZEB2 as repressors. The authors conclude that the role of RUNX1 is to activate expression of the repressor proteins, thereby leading to gene repression by an indirect mechanism. These experiments and analyses lead to important new insights into the mechanisms of regulation by RUNX1. The authors also provide some discussion about the possible application of these insights into RUNX1 and ZEB1 and 2 function in other cellular contexts.

The experiments are performed carefully and with a good replication structure. The analyses are appropriate and rigorous. The text is clear for the most part but it could be more concise.

The manuscript could be improved by addressing the following specific points.

(1) The results of the triplicate determinations are shown for RNA-seq in Fig. 2, but the patterns across replicates are only shown in the heatmaps of signal density in peaks in the Supplement. Showing at least some of the differentially bound or accessible regions (e.g. those in Fig. 3 F and G) for all three determinations would provide compelling examples of consistency in the data. Such a figure panel would be a strong addition to the Supplementary figures.

(2) The authors should state that mK4 cell line is from mouse early in the paper; the first I read of mouse as the species of origin was in the Discussion.

(3) Some important labels are obscured in "Figure 2-Supplemental Figure 1", e.g. which panels for for RUNX1 ChIP-seq and which are for RUNX2?

(4) What is the x-axis in Fig. 2C and 3C and in "Figure 2-Supplemental Figure 1B"? The label says transcription factor binding motif, but there are multiple values for each of the motifs in the color key. Are there many motifs in the RUNX class, in the AP1 class, etc?

(5) Lines 203-205: The conclusion drawn about direct repression is overly broad. The results presented do not make a case for one conclusion that applies to all repressed genes. Just using the numbers in the preceding sentence, it seems that the data DO support a model for a direct role of RUNX1 in repression of about 30% of the genes whose expression is up-regulated in the Runx1 KO. Of course, that leaves a majority of the upregulated genes as not consistent with a direct role of RUNX1, and that would be an appropriate way to describe the conclusion.

(6) In Fig. 2B, are the genomic regions with the RUNX2 ChIP-seq signal the same ones and in the same order as those shown for the RUNX1 ChIP-seq signals? Or are they for the peaks of RUNX2 binding, ordered by the strength of RUNX2 signal? These heatmaps indicate a substantial overlap in binding between RUNX1 and RUNX2, but the Venn diagram in Fig. 2D indicates that roughly half the peaks overlap. These panels need to be reconciled.

(7) The observations on lines 257-258 should be expanded. What fraction of the RUNX1-dependent ATAC-seq peaks are also bound by RUNX1? Is it a high majority? More information is presented later, but the relevant supplemental figure panels are missing, and no explicit answer is given to this question.

(8) "Figure 3-Supplemental Figure 1" appears to be missing panels A, B, and C.

(9) For GSEA (Fig. 3D), the gene sets examined need to be stated in the text or figure legend and labeled in the Figure. What is blue and what is red along the x-axis?

(10) line 280: What exactly is being referred to by the term "pioneer/activator function"? The concept of a pioneer function has evolved, and in some contexts it refers to an activity that opens previously inaccessible chromatin. Is that what is meant, and if so, how is that involved in maintenance of accessibility?

(11) lines 288-290: What aspect of the data in Fig. 3E show a "slightly decreased" RUNX2 ChIP signal at the RUNX1-dependent ATAC-seq sites in Runx1KO cells? The heat maps look the same for + and - RUNX1. The specific examples are pretty clear in panel F, but the overall pattern in panel E does not show a clear decrease.

(12) line 395: need to add "those encoding" before "several other transcription factors".

(13) lines 411-412: What data in this paper support the conclusion that RUNX1 appears to "act by making chromatin accessible to other TFs"? In order to see such an effect, one might expect an analysis of DNA sites that were previously in inaccessible chromatin, but which become accessible upon introduction of RUNX1. It is not clear how a knock-out experiment shows an effect on initiation of chromatin opening. The loss of accessibility upon RUNX1 knock out supports a role in maintaining accessibility, as the authors point out in other text.

Reviewer #3: The article by Hass et al. describes how knocking out RUNX1 in a kidney derived cell line leads to changes in chromatin organisation and gene expression even at genes that are also bound by RUNX2. They also demonstrate that lost accessible sites are enriched in AP-1 motif. In addition, they show that genes which are down-regulated are direct Runx1 targets but genes which are up-regulated are not. They propose that the up-regulation of such genes is caused by the down-regulation of ZEB repressor proteins as sites associated with such genes are enriched in ZEB motifs..

The data is of good quality and the results are interesting and if taken further, could expand upon known functions of RUNX1 in a novel cellular context.

However, I am afraid that this study is not developed far enough to support the conclusion put forward by the authors, this holds true in particular for the proposed role of ZEB proteins.

Major Comments

1. Loading controls (such as Gapdh or B-actin) must be shown for western blots in Figure 1, Figure 4 and Supplementary Figure 1.

2. Lines 183-185 and 252-255 presence of an AP-1 motif alone is not sufficient to describe these cis-regulatory regions as enhancers. Along the same vein: The loss of AP-1 motifs does not mean that this factor is binding there. To ensure that RUNX1 is required for AP-1 binding, ChIP data need to be provided

3. The fact that some down-regulated genes cannot be rescued by the presence of RUNX2 is very interesting, but again, the mechanistic basis of this finding is unclear. How does histone modification look like at these regions? Could chimeras between RUNX1 and RUNX2 be used to examine which domain of RUNX1 is required for the activation of these genes?

4. Figure 2/Supplementary Figure 2 the Venn diagram in panel D shows around a quarter of Runx2 peaks are each gained and lost following Runx1 KO. The authors mention that replicates were well correlated but this is not easy to confirm with the heatmaps shown. Here either a correlation matrix should be provided or a similar Venn diagram of peak numbers to see if the changes in response to Runx1 KO are more than expected by chance/technical variation.

5. It is puzzling to see that RUNX2 binding can be associated with inaccessible chromatin - how many of its peaks lie within open chromatin (related also to Figure 3) and what distinguishes these from the inaccessible sites?

6. Results/discussion: Lines 203-205, Figure 4. A central claim to the article is that RUNX1 is not acting as a repressor via Zeb proteins. However, the data does not support this claim. 30% of upregulated genes having a nearby RUNX1 peak still seems high even compared to the 45% of downregulated genes although a p-value is only given for downregulated genes - is this because it is also higher than expected by chance? In the absence of Hi-C data, how do they know which binding site belongs to which gene. It is known that only about 50% of all ChIP sites belong to the nearest promoter – which will create a problem with the statistical significance.

7. In the follow up experiments binding at these genes by Zeb1/2 is not documented by ChIP so there is no direct evidence that these proteins are acting as the repressors at these sites. The authors make a point that there is no antibody available, but they work with a cell line into which tagged ZEB proteins could be introduced and expressed in an inducible fashion, which would nail the issue.

8. Could RUNX1 and ZEB proteins be acting in a complex and therefore Runx1 binding is indirect given the lack of RUNX motifs? Is the t-test in Fig4 panels G & I vs the Runx1KO? If so this needs to be done against the control too to confirm if there is a full rescue - there does not appear to be, further suggesting it is not ZEB proteins alone.

9. The discussion is way too long and too speculative for the limited evidence provided.

Minor Comments

1. Introduction: It would be helpful to describe the differences between Runx1 and Runx2 so the different behaviours discussed later can be better understood.

2. Methods: Some detail is missing from the methods section making the study difficult to interpret and impossible to replicate. There is no information on RNA-seq analysis past alignment - how were differential genes called? Primer sequences and antibodies used are not provided. Data has been aligned to mm9, unless there is a good reason for this mm10 (released 2011) should be used.

3. The authors mentioning the “pioneering” function of RUNX1. However, this idea is way too simplistic. The review they cite does not show such a thing. RUNX1 when expressed can cause the opening of chromatin, but it may need other factors to do this. The authors are advised to read one of Ken Zaret’s reviews for a correct definition of pioneer factors

4. Supplementary Figure 3 panels A-C are missing.

**Have all data underlying the figures and results presented in the manuscript been provided?**

Reviewer #1: Yes

Reviewer #2: Yes

Reviewer #3: None

PLOS authors have the option to publish the peer review history of their article (what does this mean?). If published, this will include your full peer review and any attached files.

Reviewer #1: No

Reviewer #2: **Yes: **Ross Hardison

Reviewer #3: No

---

## [Decision Letter · Decision Letter 1]

3 May 2021

Dear Dr Kopan,

We are pleased to inform you that your manuscript entitled "Runx1 Shapes the Chromatin Landscape Via a Cascade of Direct and Indirect Targets" has been editorially accepted for publication in PLOS Genetics. Congratulations!

Yours sincerely,

John M. Greally, D.Med., Ph.D.

Section Editor: Epigenetics

PLOS Genetics

Gregory Barsh

Editor-in-Chief

PLOS Genetics

Comments from the reviewers (if applicable):

Reviewer's Responses to Questions

**Comments to the Authors:**

Reviewer #1: Adequate response to concerns in the initial review have been made. No remaining reservations with the manuscript.

Reviewer #2: The authors have thoroughly addressed the issues raised in the first round of reviews. They have also conducted additional experiments, such as knockouts of the Zeb1 and Zeb2 genes to further develop the story about these targets of RUNX1 serving as repressors of downstream genes.

I noticed one passage that the authors may want to clarify:

Line 267: I think that "in the Runx1-dependent sites" should be added after "motifs" to clarify the class of ATAC-seq peaks being described (unless the observed enrichment was observed in both classes).

**Have all data underlying the figures and results presented in the manuscript been provided?**

Reviewer #1: Yes

Reviewer #2: Yes

PLOS authors have the option to publish the peer review history of their article (what does this mean?). If published, this will include your full peer review and any attached files.

Reviewer #1: No

Reviewer #2: **Yes: **Ross C. Hardison

**Data Deposition**

http://datadryad.org/submit?journalID=pgenetics&manu=PGENETICS-D-20-01576R1

**Press Queries**

---

## [Editor Report · Acceptance letter]

7 Jun 2021

PGENETICS-D-20-01576R1 

Runx1 Shapes the Chromatin Landscape Via a Cascade of Direct and Indirect Targets 

Dear Dr Kopan, 

We are pleased to inform you that your manuscript entitled "Runx1 Shapes the Chromatin Landscape Via a Cascade of Direct and Indirect Targets" has been formally accepted for publication in PLOS Genetics! Your manuscript is now with our production department and you will be notified of the publication date in due course.

With kind regards,

Andrea Szabo

PLOS Genetics

On behalf of:
